# A Qualitative Study of Child and Adolescent Mental Health during the COVID-19 Pandemic in Ireland

**DOI:** 10.3390/ijerph18031062

**Published:** 2021-01-25

**Authors:** Katriona O’Sullivan, Serena Clark, Amy McGrane, Nicole Rock, Lydia Burke, Neasa Boyle, Natasha Joksimovic, Kevin Marshall

**Affiliations:** Department of Psychology, The National University of Ireland, Maynooth W23 F2H, Ireland; Serena.clark@mu.ie (S.C.); amy.mcgrane@mu.ie (A.M.); nicolerock93@yahoo.com (N.R.); lydiaburke34@yahoo.com (L.B.); neasa.boyle.2019@mumail.ie (N.B.); natasha.joksimovic.2019@mumail.ie (N.J.); kevmar@microsoft.com (K.M.)

**Keywords:** mental health, psychological harm, vulnerable groups

## Abstract

Mitigating the adverse physical health risks associated with COVID-19 has been a priority of public health incentives. Less attention has been placed on understanding the psychological factors related to the global pandemic, especially among vulnerable populations. This qualitative study sought to understand the experiences of children and adolescents during COVID-19. This study interviewed 48 families during the COVID-19 pandemic restrictions, and a national lockdown, to understand its impacts. The study used an Interpretative Phenomenological Analysis (IPA) methodology. Parents and children discussed the negative impact of the restrictions on young people’s wellbeing. Children and adolescents experienced adverse mental health effects, including feelings of social isolation, depression, anxiety, and increases in maladaptive behaviour. Families with children with Autism Spectrum Disorders reported increased mental health difficulties during this period mostly due to changes to routine. The findings highlight the impact of severe restrictions on vulnerable populations’ wellbeing and mental health outcomes, including children, adolescents, and those with Autism spectrum disorder (ASD).

## 1. Introduction

Discourse and policies around COVID-19 disproportionately focus on the adverse effects of the public health crisis on adults. Research shows that depression, anxiety, and post-traumatic stress are the most common psychological reactions to the pandemic in adults [1]; however, the impact of COVID-19 on young people is not fully understood. Though children and adolescents are not at the forefront of the pandemic, the United Nations Sustainable Development Group (UNSDG) labels them a vulnerable group that is at-risk for becoming its biggest victims and for some, the impact will be lifelong [2]. From a medical perspective, evidence suggests that young people are less likely to experience severe symptoms from contracting the disease [3] but this does not mean they are equally able or supported to cope with the psychological, economic, and social effects. Furthermore, increasing inequalities in parents’ resources and shifting conditions of home environments may exacerbate disparities in opportunities for advantaged and disadvantaged youth [4]. 

The short-term policy strategies of governments have focused on mitigating the physical risks of COVID-19 by placing limits on social interactions and freedom of movement. These policies have transformed how children aged 4–12 years, and adolescents aged 13–18 years behave in their lives. There has been an interruption to care arrangements, youth education and leisure services, as well as to schools and other organisations. Disruptions have been shown to disproportionately affect women [5], while reductions in families’ incomes, from a decrease in work hours, have negatively impacted households’ quality of life. These factors have had significant impacts on children’s health and wellbeing [6]. Reports by the UNSDG show that as of April 2020, more than 188 countries had some form of school closures, placing over 1.5 billion children and adolescents out of schools [2]. The impact of these policies on the mental health of young people is not yet clear. Though the longitudinal data is not yet available, the assumption is that there will be mental health consequences resulting from the pandemic [7]. Research shows that social isolation and loneliness are impacting young people [8]. There have been increases in adolescent anxiety [9,10], which has been due in some part to pandemic related restrictions [11,12]. The extent of the impact will likely depend on factors like age, gender, socioeconomic indicators, and country of origin [13]. The short- and long-term impacts are unclear, as well as if it will cause an increase in mental health disorders in younger people. As many mental health disorders begin in childhood, it is essential to consider children and adolescents’ psychological welfare during crises. This consideration is especially vital as young people are more susceptible to experiencing long-term consequences of mental health [14]. To date, there has been little systematic research exploring the psychological impact of the pandemic on children and adolescents in Ireland. This study, alongside relevant international research, analyses and presents essential data that informs the impact of COVID-19 on young people living in Ireland. 

### The Psychological Impact of Covid-19 on Children and Adolescents

Some groups may be more vulnerable to the psychosocial effects of the COVID-19 pandemic than others. Because they are in a critical period of development, with half of all mental health disordered developing before the age of 14 [15], children and adolescents must be provided with adequate supports. Factors associated with mitigation measures such as social distancing, family discord, school closures, fears about the future, and quarantine are disrupting young people’s lives [4,16]. These disruptions include changes in routine, break in the continuity of learning with the closure of schools, break in health care missed significant life events and the loss of a sense of security and safety [17]. Early research has indicated an increase in adverse mental health outcomes for young people. These include increased levels of depression and anxiety [9,10] as well as a greater likelihood of post-traumatic stress symptoms [14].

In Ireland, the landscape of youth mental health has deteriorated over recent years. There is a notable increase in anxiety and depression among young people and higher rates of self-harm [18]. Furthermore, Ireland has the fourth-highest suicide rate in Europe among young people [19]. With the assumption that the psychosocial effects of COVID-19 will disproportionately affect young people, the immediate and long-term consequences must be understood. This understanding will aid the development of supports and provide insight into how best to respond to crises like those seen in recent times. The expected impacts of COVID-19 on young people will likely differ depending on age and the family’s social and demographic qualities. Parental stress caused by the pandemic-related factors may have a more significant impact on younger children, which can result in behavioural problems and can have a substantial effect on children with developmental disabilities [20]. Research also shows that increases in parental stress levels during a pandemic directly interfere with a child’s quality of life [21]. This systematic review of the impact of pandemics reveals that quality and duration of sleep and decreased levels of physical and outdoor activities can, and does, prevent child development from reaching its full potential in times of crisis.

Children and adolescents with pre-existing mental health conditions may be more susceptible to new mental health conditions during the pandemic [22]. The closure of schools has led to many children losing access to mental health services received directly through their schools [23]. A study in the United States (US) found that 13.2% of adolescents were receiving some form of mental health service from their school before the pandemic [24]. By closing schools, they may have lost their support network during a time where they may be suffering from social isolation, loneliness, and anxiety as a result of COVID-19 [14]. Furthermore, research suggests that parental mental health during COVID-19 can influence child mental health [25]. Increases in parental stress during quarantine is associated with their children’s negative emotions and behaviour; the more stressed a child’s parent is, the more stressed the child themselves are [26]. These parental mental health difficulties may lead to negative parenting behaviours related to nurturing and monitoring, decreased attention to a child’s needs and more significant dysfunction within the home [27,28]. Increased family stress exacerbated by home-schooling and reduced psychological supports through school means that young people might be more vulnerable [29].

Mental health problems emerge in early childhood, with almost half of the acute mental illnesses occurring by the age of 14, and 75 per cent manifest by the mid-20s [4]. It is well documented that children and adolescents’ psychological wellbeing is critical for good mental health later in life. Mental Health America reported that the mental health impacts of COVID-19 are more pronounced in age groups under 25 and are placing a generation at risk of long-term effects if these issues are not addressed. Early evidence suggests that around 9 in 10 young people are screening with moderate-to-severe depression and 8 in 10 with moderate-to-severe anxiety, as a result of the crisis [30]. This study extends this agenda by exploring the impact that the crisis has had on the family’s wellbeing, focusing specifically on young people’s mental health. The potential short and long-term impact of COVID-19 on children and adolescents’ psychological wellbeing is unclear, making this study imperative to understanding these prospective impacts and informing needed interventions. This research uses a qualitative approach to examine children and parents’ perceptions and experiences of COVID-19; it explores how the restrictions imposed on families have impacted young people’s psychological wellbeing in Ireland.

## 2. Method

### 2.1. Participants

Forty-eight families (*N* = 94) were recruited for this study. Parents, guardians, caregivers, and their children were recruited in this study to share their experiences of COVID-19. One hundred and nine participants agreed to participate, and 94 in total completed interviews. Twenty per cent were parents on their own, and 3% were couples. 

Families were recruited through a convenience sampling method and snowballing sampling techniques. Participants were invited to participate through social media and other networking platforms. The final sample demographics include two-parent families, single-parent families, those from different socioeconomic backgrounds, and rural and urban settings. Though the researchers sought an inclusive sample set, participation was based on people’s willingness to participate in the study, and 48 families agreed to participate. Additionally, more female parents or caregivers were willing to participate (see Table 1) This study received ethical approval from The Social Research Ethics Subcommittee, Maynooth University (2407411).

### 2.2. Design

This qualitative study features an Interpretative Phenomenological Analysis (IPA) approach [31]. This study was conducted and reported in accordance with the Consolidated Criteria for Reporting Qualitative Research (COREQ). The design featured online interviews with families, and the researcher asked open-ended questions through a semistructured interview format. This design aimed to explore individual experiences of COVID-19, exploring children and adolescent mental health during this period.

The IPA interview schedule was developed based on academic expertise. The semistructured interview employed “a funnelling technique” in which the methodology allows discussion to flow from general topics to specific themes [31]. This design was administered due to its allowance to provide structure and yield rich data while maintaining consistency across interviews.

### 2.3. Procedure

Participants were recruited from social media and various networking platforms. Participants signed up online and gave their contact details for the researcher to schedule an initial briefing and to send the relevant information forms. In order to prevent the spread of COVID-19, hard copy consent forms were not used. Consent was achieved verbally through the recruitment phone call and during the interview. Obtaining consent from the participant was recorded in the interview. The interviews were conducted through Microsoft Teams, recorded, and transcribed, based on participant consent. Microsoft Teams was the chosen method for the interviews as the research team has a professional Microsoft Office account, ensuring data protection and ownership of the data. The researcher’s Microsoft Office (Redmond, WA, US) account was used to store participant details and the recorded audio from each interview.

The interviews were conducted during the first set of pandemic restrictions in Ireland. The interviews lasted for approximately 30 min and were led by the researcher. At the beginning of each interview, families were informed of their right to leave the research at any time without suffering any consequences. Each interview took place with families as a group. Following the initial briefing and obtaining consent, the researcher facilitated open-ended questions through a semistructured interview. This allowed for flexibility of questions during the interview. Families’ “funnelled” through their experiences of COVID-19 and discussed their experience of government restrictions and their shifted family dynamic. Following each interview, the researcher debriefed the family and asked if they had any questions. When the interview was finished, the researcher saved the audio file and transcribed the interview verbatim. The participants were also contacted to ensure that they were happy with their interview and if they had any queries about the research process.

Families reflected on their situation during the pandemic and the first lockdown in Ireland. At this time, people were only allowed to leave home to shop for food and exercise briefly within 2 km of their homes. All schools were closed, and people were asked to work from home where possible. Additionally, public or private gatherings were banned, as were visits to hospitals and prisons, with some exceptions on compassionate grounds. People aged over 70 and vulnerable groups were told to “cocoon”, and travel to offshore islands was limited to residents only.

### 2.4. Data Analysis

A thematic approach to IPA was used for the data analysis using the NVIVO software package (QSR, Melboure, Ausrtalia). The analysis we employed followed the six steps outlined by Smith and Shinbourne [31]:**Reading/Re-reading**—The research team familiarise themselves with the interview concept by immersing themselves with the transcript.**Coding**—The research team identify codes and organise them into initial themes.**Clustering**—Themes are emerged by common themes and subthemes.**Iteration**—The iterative process involves several revisions, including checking themes, subthemes, and quotes.**Narration**—The research theme develops a narrative based on the findings. The narration process involves describing the themes and using quotes to illustrate them.**Contextualisation**—The researchers interpret the findings within the context of existing literature.

IPA thematic analysis moves from descriptive to interpretive. The process consists of first categorising each transcript into broad themes and through continued review translating the data into more specific themes [32]. In line with the IPA approach, focuses on the subjective lived experiences of people [33], and, therefore, the researchers demonstrate reflexivity throughout the study [34]. To elaborate, reflexivity encourages the researchers to consider how their subjective worldviews may impact the research process. This approach benefits this research’s quality, particularly concerning the data analysis as it buffers against personal experiences and biases impacting the research findings [35].

The interviews were conducted by Amy McGrane, MSc, who was provided with interview training by the research’s principal investigator, Katriona O’Sullivan. Once interviews were transcribed, to ensure procedural consistency across interviews, all researchers initially separately coded the interviews. The themes that emerged were then discussed and analysed together to determine the final themes. The research also counted the frequency of occurrences of reference to each code, and these are presented in Table 2.

## 3. Results

### 3.1. Children Have Borne the Brunt of Covid-19

“My conclusion is that children have borne the brunt of this.”(Parent)

Throughout the interviews, we observed the stresses and strains placed on families in the COVID-19 crisis. One common theme that emerged was the direct impact that the restrictions were having on young people. Throughout the interviews, both parents and children referred to the extra strain placed on children during the pandemic. Many parents argued that their children struggled the most with the lockdown measures. For example, one mother described the impact on her young children, and how there was nowhere for them to go during the crisis, they did not have the benefit of being able to connect through technology:

“My conclusion is that the children have taken, have borne the brunt of this. Really, you know they didn’t…have much to go on and my biggest point and the reason I really wanted to participate in this study is because a 9-year-old and a 7-year-old are not able to talk like we’re able to talk over a zoom call. It’s just it’s just not. I don’t think they’re neurologically mature enough, I know they’re digital natives, but I don’t think they have the capacity to learn or communicate across digital platforms.”(Parent)

Children were socially withdrawn and socially isolated—and parents said that children felt it much more than adults due to their lack of fluidity with digital means of communication. Additionally, parents were concerned that children were not supported to stay connected to their friends. There was limited peer interaction throughout the lockdown: 

“Yeah, I think that kills them the most with not having any social time, not seeing their schoolmates.” (Parent) “And you know, there’s a lot of stuff that they need to talk about, and I think even nothing to do with education, but even seeing their, their classmates on a screen and being able to share experiences would have gone a long way towards alleviating any source of negative mental health.”(Parent)

A recurring theme was how the young people were missing out the most through the pandemic and that their life was changed for the worse. Especially when it came to schooling and those transitions that young people usually get to experience.

One parent describes this perfectly: “It was really sad, like they were really upset because they lost out on all the typical rites of passage that they were looking forward to, like the graduation their end of year ceremony” (Parent). Children and young people were seen as having the worst experiences in the COVID-19 crisis because they could not be children.

### 3.2. Children’s Mental Health

#### 3.2.1. Social Isolation

According to both the parents and children in this study, children’s adverse mental outcomes were provoked by experiences of loneliness and social isolation. As young as five, children stated that, “I hated everything about COVID” (Child aged 5). when discussing being separated from friends and school. All the parents interviewed illustrated deep concern over the effects of the prolonged social isolation for their children:

“Because I even found my son, he’s only gone seven since June, he was six at the time. He was a really outgoing child but because with the lock down. The hassle I had to get him back out on the road just, you know when the restrictions were lifted a bit. He went really into himself- really shy, now that has all completely turned back around, he is back to his old self now. But for a few weeks, I was actually getting worried about him like his mental health at six years of age. He didn’t want to go out and he got very into himself because he wasn’t out with friends.”(Parent)

A common understanding among parents was that their children would catch up on what they missed out academically. However, the children were most negatively affected by the lack of social interaction with their peers:

“It was the lack of social interaction with his peers. I wasn’t even worried in the beginning. I was a bit like Oh God, his maths, his whatever. But after a while I was like I don’t actually care about that. That can be picked back up well, interacting with your peers is so important you know and just weeks and months of just sitting in front of a screen.”(Parent)

Another parent described the subtle changes in their child’s behaviour and how they felt powerless to help:

“I’d say he would miss the social aspect…we have observed certain things, certain behavioural changes, not big things, but just there’s more frustration from being home all the time. I think that you just can’t. You can’t just replicate the craic a bunch of five- and six-year-olds have together and the engagement that they have…”(Parent)

#### 3.2.2. Stress over Home- School Expectations

Another catalyst of negative mental health outcomes for children was the stress placed on them to complete their home-schooling. One child stated that:

“There were so many projects, nearly there was one project every week and then there was like every single subject on the thing and like she’s getting us to do like so much. And then I just got like really like it was too much for me. ‘Cause I’m used to like smaller work and she would give us more time to finish it.”(Child aged 11)

This child found distance learning difficult and struggled to work independently. Like many other children, she missed the collaborative working style of the classroom and found home-school expectations stressful. This theme was recurrent in this study. Many families reported that to salvage the wellbeing within their family unit, they would not inflict home-schooling requirements onto their children:

“She really missed her friends and she’s at that age where she was just, she used to get quite emotional about, and anxious about getting it done and stuff like that. So there were days where I just kind of told her look. We don’t have to do anything today. You know we can just. You know, go upstairs and read a book or something, or chill out. Or watch a movie or something like that…. We will just we just put on a movie and chill out and forget about school for today.”(Parent)

#### 3.2.3. Anxiety

Several parents reported that their children experienced anxiety throughout the lockdown imposed by COVID-19. For example, one parent noticed the emotional changes in her children after a few weeks of restrictions:

“In the beginning it was a huge novelty. Of course, this is great. You can do half an hour work and watch TV and play Xbox, but as the weeks went on, I felt all of us really. The morale was very low. Our moods were very low. He was very tearful. And even the teenager who was living the dream of doing a tiny bit of work and being online all day…he was low and crying and tearful.”(Parent)

Other parents articulated that their children’s anxiety manifested in attachment issues and frustration:

“My daughter really struggled a lot, and she was like she wasn’t really able. She wanted a lot of input, even though she didn’t really need it. So, she was very, very anxious and so basically, she’d want to be sitting here. You know, when you’re doing, the work, this is the bedroom she might be sitting on the bed and you know she’d be interrupting. Yeah, and then she’d be huffing and puffing, you know what I mean.”(Parent)

Parents described having to control the information being given to their children about COVID-19 and how watching the news caused some anxiety.

“She kept asking to watch the news, I said no, you’re not watching the news. I’ll explain to you what’s going on afterwards.”(Parent)

Another parent described the anxiety, which emerged when her child was considering the restrictions being placed on the family:

“she asked…if we go on a bit further because the lady and the radio says that you’re not going to go more than two kilometers? And I’m gonna go out, I’m going to be. Are we going out about 2 km zone here? We ought not be doing this, you know, yeah, so there’s no, you can’t sugarcoat if the child is able to be picking up on public health messaging.”(Parent)

Overall families described their children as anxious and high strung during the pandemic, and that this was facilitated by the media coverage of the pandemic and the lack of access to friends and family.

#### 3.2.4. Negative Behavioral Changes

Interestingly, many parents of children younger than 10, noticed that their children developed increased negative or maladaptive behavioural changes following the restrictions. For example, one single parent noticed that her son reverted to an old bedwetting behaviour, possibly correlated with not seeing his father for an extended period:

“For a good few weeks, he didn’t see the kids (their father), so that did affect my youngest son. One of the twins, his behaviour went all over the place and he was acting up and everything and went backwards and he was starting to bed wet again and still is, so I had all like to deal with as well.”(Parent)

Moreover, parents reported increased frustration among their children:

“You know he’s an absolute nightmare, or beating up his sister, one or the other. I was on many calls where, like my boss would say, do you want to go and sort that out because you could hear the fighting from two rooms away.”(Parent)

However, the most reported maladaptive behaviour among young children was the decreased attachment security, and “clinginess” children demonstrated towards their parents:

“Because of the lock down he didn’t see even his grandparents. Or you know anyone else for about three months, so he got really, really clingy with my wife, like his mam. And then he couldn’t like, she has to sleep in the same room with him, and if she is going to the toilet, he would follow her. I think the lock down really did play; I mean he was pretty attached but it got way worse. We had gotten used to the behaviour, and now that were coming out of the lock down, we can see already, he’s a little bit less, because he’s getting to see his grandparents once a week and he gets to go out on more trips without being confined. So, he’s coming out of it a little bit, but there were definitely behavioural issues because of the lockdown. It was it was blowing over like the you know, but through this winter again I’d say he’d regress all over again.”(Parent)

### 3.3. Adolescents Mental Health

While adolescents experienced similar negative mental health outcomes to younger children, their mental health experiences manifested differently. Depression and anxiety were commonly reported experiences during the lockdown for adolescents, and many parents sought mental health services for their teenage children:

“He was getting very depressed. At one stage he even asked me to get a counsellor. That’s how depressed he was getting. So, I was getting really worried about it. He seems to be coming back to himself but for a while, yeah. It had more impact on him then it did the little ones.”(Parent)

Negative mental health outcomes were reported following the mourning of milestones that were cancelled due to COVID-19:

“He spends a lot of time in my mother’s house so yeah, he was pretty much locked in and he was getting very down some days. So, we were glad when they lifted a little bit that you could start going out and mixing with his friends and that a bit more. But I know that things like his graduation. He was very down about that like I think he was more down about that then missing his Leaving Cert I’ll be honest with you.”

Another parent said:

“She was robbed of a rite of passage I suppose. And she does feel upset over that and disappointed.”(Parent)

In addition, many parents reported that teenagers experience increased anxiety over future transitions, such as the transition into secondary school:

“So she had months on her own dwelling on that. So that wasn’t a good year for her, and that’s a funny age as well, because you’re kind of like maturing and then puberty and everything. Yeah, full on hormones and she doesn’t like change. The biggest change of her life is coming, and then she just sat at home for months to think about it. Yeah, so that that was. I had to send her to a counsellor then after that ‘cause she just was struggling with anxiety for it”.(Parent)

The confined restrictions associated with COVID-19 had adverse mental health outcomes for all adolescents interviewed in this study. The lack of freedom forced teenagers to be confined to their household, which provoked various challenges. For example, one teenager stated that:

“we stressed out so much because my parents are actually in the middle of a divorce, but they live in the same house. So that alone is stressful, then lockdown came, and my dad was always at home. Obviously, we couldn’t go anywhere we can’t get out of the house and it really added to the stress ‘cause you can’t walk into a room where my two parents are because you’re like stepping on egg-shells. It’s not really a relationship. I know for me it really effects like my mental health and everything. I hate being stuck inside. I don’t mind being inside, but you know, I just wanted my one time where I was like. I really want to go out. I really wanna get outta here. I even missed getting the bus.”(Child)

For this teenager, she missed everyday life outlets that would typically distract her from a difficult family situation. Many teenagers suggested that they struggled with the confinements associated with lockdown despite their family dynamic. Moreover, adolescent parents exhibited concern regarding their lack of routine within the family household:

“it was a struggle to get my 14-year-old out of the bed, because, you know, they’re not tired enough, so they’re not going to sleep. First of all, it was midnight. Then it was 1 AM like the average now its 3 AM for the two older ones. And of course, they don’t get up till 11 or 12.”(Parent)

This disruption in routine has been heavily correlated with an increase in screen time, including staying up to play video games or the influx of social media usage:

“then for the past three months she’s lay on the sofa on her phone, and there’s nothing I can do that will motivate her to do anything. You know, should I take the phone off her? Should I ban her?”(Parent)

The removal of structure, routine, and outlets of support for teenagers during COVID-19 has been associated with the accentuation of their modern-day challenges. Parents articulated that their concerns for their teenage children have multiplied since they lost their standard outlets of support. For example, one parent discusses an increased awareness of her daughter’s body image issues and anxiety during this time:

“We were on holiday last year with family that live in the UK and their daughter sent her some pictures that had come up, like they were looking at from last year and my daughter was just like Oh my gosh like, I hate myself. I hate myself. I hate my body and it was purely body image and she’s hitting that age.”(Parent)

There was reference made to some of the strategies that families used to mitigate the stress and anxiety associated with the pandemic restrictions. Families described spending time outside in nature together, while baking and household chores were described as calming. Many talked about the importance of routine with some saying it helped focus them.

“So, we had to go and buy a printer so he could print out worksheets and stuff for him to do and he painted a lot of furniture. He painted the decking. He did gardening… You know anything just to keep him busy.”(Parent)

### 3.4. Mental Health Challenges for Children and Adolescents with Autism

A theme that emerged from the analysis was the impact of COVID-19 restrictions on children with developmental disorders. Six parents, two special needs teachers, described the effect that the restrictions had on children’s mental health outcomes and students with Autism spectrum disorder (ASD) during the COVID-19 restrictions. While the children and adolescents with Autism vary in their developmental trajectories, they described similarly concerning responses to the lockdown. All participants in this study discussed how the drastic change in routine had provoked anxiety among the children with Autism:

“Especially my younger daughter is on the ASD spectrum and she will have to integrate again. She does it every time she’s in school and they’re off on Friday and back on Monday, it’s a big stress for her, so you can imagine how 6 months off is going to be for her. It’s going to be like the first day of school again.”(Parent)

Anxiety was a prominent theme among these children as all families articulated its presence in the household. A parent of an ASD child with high needs discussed that his child’s anxiety manifested in relation to his attachment difficulties:

“Like he used to freak out if my wife would leave the house at all without him, because she hadn’t left the house without him in months. He lost his routine and then the new routine again was really unhealthy in a way, because there was no, kind of external anything for him.”(Parent)

A parent and teacher who works with a class of high- functioning adolescents on the ASD spectrum articulated that his students had a comprehensive understanding of the virus which provoked fear of COVID-19:

“You know it’s funny because some of the kids that I have like they would be very tuned in to, um, just you know they, they have a special subject like most kids with Autism have special interests…. But yeah, some of them are really aware of, of these viruses and how they could lead to other viruses and they were really stressed.”(Parent)

Moreover, this awareness of COVID-19 provoked maladaptive existential anxiety categorised by an overwhelming fear of the virus:

“He said that like even if there was a vaccination, he wouldn’t get it. And, because he’s afraid of needles, and he’s afraid. Basically, he started to think an awful lot about you know what will happen to him when his parents are dead and when he’s like an individual living in the world with Autism and not having the support he’s had all his life, so he’s gone really into existential anxiety and. You know this is what COVID has done. Like you know, it’s a psychological disease as well.”(Parent)

## 4. Discussion

This research explores Irish children and adolescents’ experiences during COVID-19, specifically the pandemic’s psychological impact on this age group. The study uses an inductive qualitative research design and documents the psychological effects on young people primarily through parents and caregivers’ observation. The key findings suggest that public health restrictions had adverse implications on children and adolescents’ mental health. Parents and caregivers reported higher levels of stress, depression, and anxiety among their dependents resulting from social isolation. These conditions were exacerbated in children and adolescents with developmental disorders. These findings are concerning to healthy development because adverse psychological experiences in childhood are associated with an increased risk of anxiety later in life [36]. The themes found in Table 2 are discussed below and often intertwine.

### 4.1. Social Isolation

Research has shown that social isolation and loneliness are impacting young people [8], and there have been increases in adolescent anxiety [9,10], due in some part to pandemic related restrictions [11,12]. Families in this study also described increases in adolescent anxiety, depression, and despair. At the same time, many mourned the cancellation of milestone events and were negatively affected by the loss of routine and guidance. These findings are in line with previous research [14,29,37], which indicates that the restrictions associated with the COVID-19 pandemic have impacted the psychosocial wellbeing of young people. The research found that the use of social distancing and stay-at-home orders to mitigate the spread of COVID-19 is negatively affecting the mental health of young people as has been seen elsewhere [8]. There is an increased risk of developing psychiatric disorders, such as depression and anxiety during these formable years with half of all mental health disordered developing before the age of 14 [15]. For adolescents, this risk is complicated by how hormonal and neurobiological changes correspond to increased emotional reactivity and continual development of stress regulation and coping strategies [38]. Additionally, social interactions with peers contribute to adolescents’ social health, creating a sense of belonging and reducing feelings of burdensomeness on others. These factors are essential for interpersonal needs [38]. It is apparent from this research that as the pandemic subsides, and there is a lifting of restrictions, the emotional consequences of COVID-19 on young people needs consideration and support systems emplaced to mitigate any long-lasting psychosocial effects [9,10].

### 4.2. Stress over Home-Schooling

This research shows that the stress over home-schooling was an added burden to families. The closure of schools shifted education from the classroom to the home. This change was unprecedented, and families had little or no experience of protracted home-schooling [39]. School routines are crucial coping mechanisms, particularly for young people with mental health issues [15]. Prolonged quarantine, fear of infections, boredom, insufficient personal space, and separation from classmates and teachers causes stress in children and adolescents [12]. With home-schooling increasing family stress [29] especially in already burdened families [40], this research shows a need to support young people to engage with education in ways that are not stressful, which suit their family situation. As discussed, social isolation is damaging to the psychological wellbeing of young people. In the United States, epidemiological data indicated that 35% of adolescents who used mental health services between 2012 and 2015 availed their school’s mental health services [10]. With schools closed and the decrease in services offered by them, home-schooling stresses may be acting to exacerbate the long-term psychological impacts on young people; young people’s experience in this study point towards this.

### 4.3. Behaviour Changes, Depression, and Anxiety among Young People

The restrictions associated with COVID-19 had negative mental health outcomes for adolescents in this study. Increased levels of anxiety and heightened fear were observed. These findings matched those from previous research, where increases in adolescent anxiety have been seen [9,10]. The lack of freedom forced upon young people through the national restrictions provoked various challenges; with the closure of schools and home-quarantine increasing anxiety and loneliness among adolescents and increasing negative behaviour in younger children. Moreover, reported elsewhere, this study supports the emerging view that pandemic related restrictions [11,12] have negatively impacted the psychological wellbeing of young people.

### 4.4. Young People with Autism Mental Health Negatively Impacted

The research also observed that children and adolescents with ASD had specific mental health challenges related to the disruption to routine, which emerged as a direct result of the restrictions. These results highlight children’s mental and emotional vulnerability during this pandemic, especially those with developmental disabilities. Previous research shows that these children are at increased risk as the pandemic dissipates and that they require specific interventions to minimise having disproportionate consequences on their psychosocial and emotional wellbeing [41]. The outcomes of this research support this and indicate in the relatively small sample of families with a child with ASD; they required support to reinstate routine and support a transition back to pre-COVID times.

### 4.5. Limitations

Though this study produced valuable data, there are some limitations. First, the researchers sought a sample set that would be equally represented across specific demographics, such as single-parent families and two-parent families, socioeconomic backgrounds, and rural and urban settings. However, the sample set was dependent on the willingness of families to participate. As a result, the sample set was not equally distributed across these demographics. For instance, because Dublin has the largest population size in Ireland, many families resided in Dublin, an urban setting. Gender also played a role, as mothers were more likely than fathers to be interviewed. A second limitation is the use of the remote interview technique. Though this was an essential tool because of COVID-19, this technique limits nonverbal communication between the interviewer and interviewee and nonverbal communication can provide important insights into the interview.

## 5. Conclusions

This study, alongside international research, illustrates the mental health challenges arising from COVID-19 pandemic related restrictions. Young people are at risk of suffering psychological consequences, and we are not yet sure of the long-term impact that this will have. In addressing these difficulties, there is a growing need to implement policies that will help children and adolescents cope with the short-term and long-term psychological effects of the pandemic, especially those who are deemed more vulnerable in this group. We need to be considering the impact of further restrictions and ensure that mental health services for young people are easily accessible if we are to prevent longer-term mental health impacts [16]. With the increasing anxiety and depression levels reported among young people and the parents’ observations in this study that children are struggling to cope, it is crucial to ensure these support services are made available. Young people need support to develop healthy coping mechanisms as they begin to process the potentially adverse effects of COVID-19.

## 6. Interview Protocol

Semistructured Interview Schedule

Can you tell me a bit about your day-to-day family life during the COVID-19 pandemic?Can you tell me a little bit about what your average school day looked like before the COVID-19 pandemic?Did you do homework?Favourite subjects?What was fun in school?Do parents work—how does school life and home-life meet?Can you talk about how the COVID-19 has affected your schoolwork?Did the school talk to you about what was happening at the start?Did you (the child) know what was happening?Was your work life affected?Were there any emotional effects?Were you able to engage with school subjects and content?Who in the household manages the home-schooling?Many families have found home-schooling difficult or strange—some have done it, some have not—can you tell me if you have been able to keep up with school during the crisis? Tell me how your day-to-day school engagement looks (even if there is none it is fine).Is there anything that has been particularly helpful to you over the last few months in doing home-school work?Is there anything that has been particularly challenging?How have you had contact with school?Would you have preferred different contact?How about you (child) did you see your teacher in the last while?Have you any ideas on how we could have made this better for children?Do you guys use computers tablets and/or the Internet—tell me what you like to do on these?Do you use these for school-work? Is there any apps or sites you have used for home schooling?Can you describe what that has been like?Any challenges?Any good points?Does this feel the same as school learning?Have you engaged with Raidió Teilifís Éireann (RTE) school or any specific things that support school work?What was it like? Can you talk me through your answer a little?If this pandemic happened again what do you think schools could do to help you?Is there anything that we could do online to help with schooling in the future?Would you like to tell me anything important about your home school experience that you have not had a chance to say or has just occurred to you?Are you worried about going back to school?What is the biggest thing you both/all miss about going to school? What don’t you miss?Can you talk about how the COVID-19 has affected your family?Can you tell me what you think the future will look like for you and your family?

## Figures and Tables

**Table 1 ijerph-18-01062-t001:** Demographics of the sample set by gender.

Interviewee Classification	Female	Male
Parent	43	6
Child	21	24

**Table 2 ijerph-18-01062-t002:** The referral and percentage of qualitative themes in this data set.

Theme	Referral (out of 48)	Percentage
Children Have Borne the Brunt of Covid-19	24	53%
Social Isolation	36	80%
Stress over Home-schooling	31	69%
Negative Behavioural Changes	16	36%
Difficulty being confined in the Household	21	47%
Depression/Anxiety among Young People	13	29%
Young People with Autism Mental Health Negatively Impacted	6	11%

## Data Availability

The data that support the findings of this study are available on request from the corresponding author. The data are not publicly available due to privacy or ethical restrictions.

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
