# Peer review of "A Qualitative Study of Child and Adolescent Mental Health during the COVID-19 Pandemic in Ireland"

_ijerph, 2021, doi:10.3390/ijerph18031062_

Round 1

Reviewer 1 Report

Dear authors, 

You have a nice study. Background literature is severely lacking in my opinion, but other than that the study looks solid. My feedback for needed edits, and suggested edits is below.

Suggestions for needed change to improve clarity:

Some more detail regarding what the current situation was for your participants during the pandemic would be very useful. Were there any government 'social distancing' policies active at the time of the study? When were the interviews conducting? Were they all conducted within a single month (if so, what month?), or were they spread out over a longer time frame? Were the government policies around social distancing the same (or highly similar) across respondents? Or was it fairly heterogenous? From the qualitative write up it sounds like schools were closed at the time of the study? Or were participants reflecting on past school closures?

I can't see appendix B... were participants reflecting on their current experience, or past experience? This is an important piece of the method that should be communicated for other researchers to be better able to interpret your study (it would be useful to make this clear in-text regardless of the appendix).

p7. lines 287-289. typos in sentence: Overall families described their children as anxiuos and highlu strung during the pandemic, and that this was facilitated by the negative meadia coverage and te lack of access to friends and family."

Suggestions for you to take or leave as you see fit to improve the paper:

Line 36. Statement "From a medical perspective, evidence suggests..." a relevant reference here: Verity et al. (2020). Estimates of the severity of coronavrus disease 2019: a model-based analysis. The Lancet Infectious Diseases, 20(6), 669-677. https://doi.org/10.1016/S1473-3099(20)30243-7

I found the finding/suggestion that children were finding the use of technology-mediated communication less useful than adults interesting. See a study that talks about technology-mediated communication use by adults during covid-19: Rogers, S. L., & Cruickshank, T. (2020, September 11). Change in mental and physical health, and social relationships, during highly restrictive lockdown in the COVID-19 pandemic: Evidence from Australia. https://doi.org/10.31234/osf.io/zutav

The introduction and discussion do not appear to comprehensively introduce/discuss the current literature examining mental health in children/adolescents during the covid-19 pandemic. I understand that your study is not a review paper. However, in present form, it is not entirely clear from the manuscript how your work really contributes to the literature. In my opinion the discussion section would have been more impressive if you had a sub-heading for each theme that is presented in Table 1 and under each sub-heading specifically discussed your finding/s in the context of relevant literature.

For example: 'Social isolation', some references:

Oosterhoff, B., Palmer, C. A., Wilson, J., & Shook, N. (2020). Adolescents’ motivations to engage in social distancing during the COVID-19 pandemic: Associations with mental and social health. Journal of Adolescent Health.

Loades, M. E., Chatburn, E., Higson-Sweeney, N., Reynolds, S., Shafran, R., Brigden, A., ... & Crawley, E. (2020). Rapid Systematic Review: The Impact of Social Isolation and Loneliness on the Mental Health of Children and Adolescents in the Context of COVID-19. Journal of the American Academy of Child & Adolescent Psychiatry.

Another example: 'Anxiety', some references:

Smirni, P., Lavanco, G., & Smirni, D. (2020). Anxiety in Older Adolescents at the Time of COVID-19. Journal of Clinical Medicine9(10), 3064.

Qi, H., Liu, R., Chen, X., Yuan, X. F., Li, Y. Q., Huang, H. H., ... & Wang, G. (2020). Prevalence of anxiety and associated factors for Chinese adolescents during the COVID‐19 outbreak. Psychiatry and Clinical Neurosciences.

Kılınçel, Ş., Kılınçel, O., Muratdağı, G., Aydın, A., & Usta, M. B. (2020). Factors affecting the anxiety levels of adolescents in home‐quarantine during COVID‐19 pandemic in Turkey. AsiaPacific Psychiatry, e12406.

Guessoum, S. B., Lachal, J., Radjack, R., Carretier, E., Minassian, S., Benoit, L., & Moro, M. R. (2020). Adolescent psychiatric disorders during the COVID-19 pandemic and lockdown. Psychiatry Research, 113264.

Another example: 'home schooling', some references:

Ewing, L. A., & Vu, H. Q. (2020). <? covid19?> Navigating ‘Home Schooling’during COVID-19: Australian public response on Twitter. Media International Australia, 1329878X20956409.

Bubb, S., & Jones, M. A. (2020). Learning from the COVID-19 home-schooling experience: Listening to pupils, parents/carers and teachers. Improving Schools23(3), 209-222.

Rinse and repeat for your other themes...

I understand the above would be a fair bit of work. I suggest it simply as an 'optional' thing for you to consider. Beefing up your discussion would help you to garner more future citations on your paper, so you would benefit from that work, but it is up to you.

One final thing I found curious is that the paper does not seem to mention any of the possible 'positive' aspects during lockdown. For example, some parents/children being brought closer together during an anxiety-provoking time. This did not happen at all in your sample? If not, this is curious, and might be worth discussing briefly?

Author Response

Suggestions for needed change to improve clarity:

  1. Some more detail regarding what the current situation was for your participants during the pandemic would be very useful. Were there any government 'social distancing' policies active at the time of the study?
    1. Inserted line 197-209 - a description of the restrictions – All families were interviewed between March 12-May 18thand were reflecting on their current situation as this was when the most extreme restrictions were in place. At this time people were only allowed to leave home to shop for food and exercise briefly within 2km of their homes. All schools were closed and people were asked to work from home where possible. All public or private gatherings were banned, as were visits to hospitals and prisons, with some exceptions on compassionate grounds. People aged over 70 and vulnerable groups were told to “cocoon”, and travel to offshore islands was limited to residents only.

  1. When were the interviews conducting? Were they all conducted within a single month (if so, what month?), or were they spread out over a longer time frame?
    1. Added into line 197-209
  2. Were the government policies around social distancing the same (or highly similar) across respondents? Or was it fairly heterogenous?
    1. Added into line 197-209
  3. From the qualitative write up it sounds like schools were closed at the time of the study? Or were participants reflecting on past school closures?
    1. Added into line 197-209
  4. I can't see appendix B... were participants reflecting on their current experience, or past experience? This is an important piece of the method that should be communicated for other researchers to be better able to interpret your study
    1. Removed reference to appendix and added to line 197-209
  5. lines 287-289. typos in sentence: Overall families described their children as anxiuosand highlu strung during the pandemic, and that this was facilitated by the negative meadiacoverage and te lack of access to friends and family."
    1. Changed and tracked line 338-340

Suggestions for you to take or leave as you see fit to improve the paper:

  1. Line 36. Statement "From a medical perspective, evidence suggests..." a relevant reference here: Verity et al. (2020). Estimates of the severity of coronavirus disease 2019: a model-based analysis. The Lancet Infectious Diseases, 20(6), 669-677. https://doi.org/10.1016/S1473-3099(20)30243-7
  2. Line 37 now includes this reference
  3. I found the finding/suggestion that children were finding the use of technology-mediated communication less useful than adults interesting. See a study that talks about technology-mediated communication use by adults during covid-19: Rogers, S. L., & Cruickshank, T. (2020, September 11). Change in mental and physical health, and social relationships, during highly restrictive lockdown in the COVID-19 pandemic: Evidence from Australia. https://doi.org/10.31234/osf.io/zutav
    1. We did not incorporate this suggestion as it was less relevant to the mental health elements
  4. The introduction and discussion do not appear to comprehensively introduce/discuss the current literature examining mental health in children/adolescents during the covid-19 pandemic. I understand that your study is not a review paper. However, in present form, it is not entirely clear from the manuscript how your work really contributes to the literature.
    1. We have added more research and introduction to the introduction (lines 50 – 67 inclusive introduction) and discussion (Lines 535-545
  5. In my opinion the discussion section would have been more impressive if you had a sub-heading for each theme that is presented in Table 1 and under each sub-heading specifically discussed your finding/s in the context of relevant literature.
    1. We have rewritten the discussion and included subheadings and related this to the literature- lines 475-556
  6. For example: 'Social isolation', some references: Oosterhoff, B., Palmer, C. A., Wilson, J., & Shook, N. (2020). Adolescents’ motivations to engage in social distancing during the COVID-19 pandemic: Associations with mental and social health. Journal of Adolescent Health.
    1. Added line 499

Loades, M. E., Chatburn, E., Higson-Sweeney, N., Reynolds, S., Shafran, R., Brigden, A., ... & Crawley, E. (2020). Rapid Systematic Review: The Impact of Social Isolation and Loneliness on the Mental Health of Children and Adolescents in the Context of COVID-19. Journal of the American Academy of Child & Adolescent Psychiatry.

  1. Added line 489

  1. Another example: 'Anxiety', some references:

Smirni, P., Lavanco, G., & Smirni, D. (2020). Anxiety in Older Adolescents at the Time of COVID-19. Journal of Clinical Medicine9(10), 3064.

Qi, H., Liu, R., Chen, X., Yuan, X. F., Li, Y. Q., Huang, H. H., ... & Wang, G. (2020). Prevalence of anxiety and associated factors for Chinese adolescents during the COVID‐19 outbreak. Psychiatry and Clinical Neurosciences.

Kılınçel, Ş., Kılınçel, O., Muratdağı, G., Aydın, A., & Usta, M. B. (2020). Factors affecting the anxiety levels of adolescents in home‐quarantine during COVID‐19 pandemic in Turkey. AsiaPacific Psychiatry, e12406.

Guessoum, S. B., Lachal, J., Radjack, R., Carretier, E., Minassian, S., Benoit, L., & Moro, M. R. (2020). Adolescent psychiatric disorders during the COVID-19 pandemic and lockdown. Psychiatry Research, 113264.

  1. Added lines 60-63, and 490-491

  1. Another example: 'home schooling', some references:

Added - Ewing, L. A., & Vu, H. Q. (2020). <? covid19?> Navigating ‘Home Schooling’during COVID-19: Australian public response on Twitter. Media International Australia, 1329878X20956409.

Bubb, S., & Jones, M. A. (2020). Learning from the COVID-19 home-schooling experience: Listening to pupils, parents/carers and teachers. Improving Schools23(3), 209-222.

  1. Added Lines 128-133 and 524-525
  2. Beefing up your discussion would help you to garner more future citations on your paper, so you would benefit from that work, but it is up to you.
    1. Completely re-written lines 475-556
  3. One final thing I found curious is that the paper does not seem to mention any of the possible 'positive' aspects during lockdown. For example, some parents/children being brought closer together during an anxiety-provoking time. This did not happen at all in your sample? If not, this is curious, and might be worth discussing briefly?
    1. Discussed briefly in line 428-435

Reviewer 2 Report

I found it a really interesting article.

The characteristics of the final sample should be further described (e.g., single-parent families, interviews with mothers vs. fathers, socioeconomic level, urban vs. rural...). Some reflection could be added on how the sampling procedure may have affected the sample profile.

Reference is made to two appendices, but I have not found them. It is important to ensure that they are available in the publication or, if not, to remove the reference to the appendices.

Some recommendations that could be taken into account if the authors consider it appropriate:

The explanation of the fulfilment of ethical criteria in research could be extended. Some are pointed out, but the possibility offered to families to leave the research at any time, without suffering any consequences, could be added.

I would delete the title of line 339 ("Difficulty being Confined in the Household"), I think it is not necessary since the previous paragraphs also relate to that issue, and there are no more sections.

In table 1, two new columns could be added, differentiating the frequencies in the group of children vs. adolescents.

Perhaps the strategies for adaptation to the situation by children and adolescents could have been described more. Knowing the strategies with which people have tried to adapt to the situation can help to better understand the negative consequences of the situation.

Perhaps the description of the evolution of the situation over the months could be extended (as in lines 266 or 292) or if some typical sequence of situations has been detected.

Author Response

  1. The characteristics of the final sample should be further described (e.g., single-parent families, interviews with mothers vs. fathers, socioeconomic level, urban vs. rural...).
    1. Lines 155-159
  2. Some reflection could be added on how the sampling procedure may have affected the sample profile.
  • These points have been addressed in the paper in section ‘participants’lines 155-159 and in limitations section lines 571-583
  1. Reference is made to two appendices, but I have not found them. It is important to ensure that they are available in the publication or, if not, to remove the reference to the appendices.
    1. Removed reference to these

Some recommendations that could be taken into account if the authors consider it appropriate:

  1. The explanation of the fulfilment of ethical criteria in research could be extended. Some are pointed out, but the possibility offered to families to leave the research at any time, without suffering any consequences, could be added.
  • This point is now reflected in the paper in section ‘procedure’ lines 187-189
  1. I would delete the title of line 339 ("Difficulty being Confined in the Household"), I think it is not necessary since the previous paragraphs also relate to that issue, and there are no more sections.
    1. Deleted
  2. In table 1, two new columns could be added, differentiating the frequencies in the group of children vs. adolescents.
    1. While this is an interesting point we felt that by differentiating the groups this would overly complicate the table
  3. Perhaps the strategies for adaptation to the situation by children and adolescents could have been described more. Knowing the strategies with which people have tried to adapt to the situation can help to better understand the negative consequences of the situation.
    1. Included lines 428 to 435
  4. Perhaps the description of the evolution of the situation over the months could be extended (as in lines 266 or 292) or if some typical sequence of situations has been detected.
    1. In the method section we have included an overview of the restrictions in Ireland to provide an idea of the sequence of restrictions lines 197-209

Reviewer 3 Report

This manuscript entitled "A Qualitative Study of Child and Adolescent Mental Health During the COVID-19 Pandemic in Ireland" aimed to understand the experiences of children and adolescents during COVID- 19.

The manuscript is very interesting. However, some issues should be addressed by the authors:

ABSTRACT

  • Please, remove the ASD acronym and use the full expression.
  • Please, include more information about the data analysis. How was the interview analyzed?

INTRODUCTION

  • It is important for the background to include recent references about mental health prevalence from just before the pandemic and other results from mental health during the pandemic.

METHODS

  • What was the reason for selecting 45 families? How was this amount selected?
  • It is not clear enough how was the interview conducted. It was a big group or individual? How was the order to answering the question? How long with each participant? Much more should be explain about how was performed the inteview.

RESULTS

  • Table 1 is not mentioned in the result paragraphs.

DISCUSSION

  • Limitations should be included as well as the strong points.
  • Conclusion in the last paragraph is not clear and should be rewritten.

REFERENCES

  • Several recent articles from IJERPH could be cited.
  • Below I suggest some articles pre and during COVID pandemic about mental health in adolescents which may improve the introduction and discussion sections:

Escobar, D.F.S.S.; Noll, P.R.S.; Jesus, T.F.; Noll, M. Assessing the Mental Health of Brazilian Students Involved in Risky Behaviors. Int. J. Environ. Res. Public Health 202017, 3647.

Seven, Ü.S.; Stoll, M.; Dubbert, D.; Kohls, C.; Werner, P.; Kalbe, E. Perception, Attitudes, and Experiences Regarding Mental Health Problems and Web Based Mental Health Information Amongst Young People with and without Migration Background in Germany. A Qualitative Study. Int. J. Environ. Res. Public Health 202118, 81.

Tahara, M.; Mashizume, Y.; Takahashi, K. Coping Mechanisms: Exploring Strategies Utilized by Japanese Healthcare Workers to Reduce Stress and Improve Mental Health during the COVID-19 Pandemic. Int. J. Environ. Res. Public Health 202118, 131.

Escobar, D.F.S.S.; Jesus, T.F.; Noll, P.R.S.; Noll, M. Family and School Context: Effects on the Mental Health of Brazilian Students. Int. J. Environ. Res. Public Health 202017, 6042.

Author Response

ABSTRACT

  1. Please, remove the ASD acronym and use the full expression.
    1. Removed
  1. Please, include more information about the data analysis. How was the interview analysed?
  1. Included in line 18-19

INTRODUCTION

  1. It is important for the background to include recent references about mental health prevalence from just before the pandemic and other results from mental health during the pandemic.
    1. Some new references have been added throughout – here is the new written section which we inserted “Though the longitudinal data is not yet available, the assumption is that there will be mental health consequences resulting from the pandemic (Daly and Robinson, 2020). Research thus far shows that social isolation and loneliness are impacting on young people (Loades et al., 2020), and there have been increases in adolescent anxiety (Qi et al., 2020; Smirni, Lavanco, & Smirni, 2020) which has been due to pandemic related restrictions (Kılınçel, Kılınçel, Muratdağı, Aydın, & Usta, 2020; Guessoum et al., 2020).”
    2. See lines 50-63

METHODS

  1. What was the reason for selecting 45 families? How was this amount selected?
    • Paper now reflects that 45 families were selected because participation was based on willingness of families to be involved and in the end 45 chose to take part
    • See line 155-159 and limitation section
  1. It is not clear enough how was the interview conducted. It was a big group or individual? How was the order to answering the question? How long with each participant? Much more should be explain about how was performed the interview.
  • Clarified in the paper that families were interviewed separately line 187-190
  • The paper now reflects that the interviews were semi structured to allow for flexibility in questions/answers
  • The paper indicates 30 minutes were given

RESULTS

  1. Table 1 is not mentioned in the result paragraphs.
    • We have added in the method a line which says “The research also counted the frequency of occurrences of reference to each code and these are presented in table 1” see line 225-226

DISCUSSION

  1. Limitations should be included as well as the strong points.
    1. Included limitation section see line 572-584
  1. Conclusion in the last paragraph is not clear and should be rewritten.
  1. Re-written see lines 559-571

REFERENCES

  1. Several recent articles from IJERPH could be cited.
    1. Done
  1. Below I suggest some articles pre and during COVID pandemic about mental health in adolescents which may improve the introduction and discussion sections:

Added to discussion - Escobar, D.F.S.S.; Noll, P.R.S.; Jesus, T.F.; Noll, M. Assessing the Mental Health of Brazilian Students Involved in Risky Behaviors. Int. J. Environ. Res. Public Health 202017, 3647.

            Included

Added to introduction - Seven, Ü.S.; Stoll, M.; Dubbert, D.; Kohls, C.; Werner, P.; Kalbe, E. Perception, Attitudes, and Experiences Regarding Mental Health Problems and Web Based Mental Health Information Amongst Young People with and without Migration Background in Germany. A Qualitative Study. Int. J. Environ. Res. Public Health 202118, 81.

            Included

Tahara, M.; Mashizume, Y.; Takahashi, K. Coping Mechanisms: Exploring Strategies Utilized by Japanese Healthcare Workers to Reduce Stress and Improve Mental Health during the COVID-19 Pandemic. Int. J. Environ. Res. Public Health 202118, 131.

Escobar, D.F.S.S.; Jesus, T.F.; Noll, P.R.S.; Noll, M. Family and School Context: Effects on the Mental Health of Brazilian Students. Int. J. Environ. Res. Public Health 202017, 6042.

  • Included

Reviewer 4 Report

This article discusses child and adolescent health during the COVID-19 pandemic.  It is clearly written and articulated. 

I feel that there are some things missing that could strengthen the paper further.  For example, the limitations section is not clear.  I also have minor suggestions as noted below:

  1. Line 16 indicates n=45 families, but how many participants were interviewed exactly?
  2. Line 38 – economic is split up into two words for some reason.
  3. Line 44: what are the defined age groups of children and adolescents that you have accepted from the literature? E.g. school aged children were defined as 4 to 12 years, and adolescents as 13 to 19 years?
  4. Line 45 mentions disruption to care arrangements as a result of the pandemic. There are also disruptions to the burden of this care, for example on women. There are also disruptions to families’ incomes, as many people have been laid off, their work hours cut, or are expected to work from home, and these are significant disruptions to the family life, and the quality of life of households.   It would have significant impacts on the health and wellbeing of children, too. It may be worthwhile to cite more information from: Syed, I.U. and Ahmad, F. (2020).  COVID19 and healthcare workers’ struggles in long term care homes. Special Issue: Health of frontline workers during the COVID-19 pandemic. The Journal of Concurrent Disorders. Available from:   https://concurrentdisorders.ca/2020/11/08/covid-19-and-healthcare-workers-struggles-in-long-term-care-homes/   

It may also be worthwhile to cite the work of Kate Power: The COVID-19 pandemic has increased the care burden of women and families.  Sustainability: Science, Practice and Policy, Available from: https://www.tandfonline.com/doi/full/10.1080/15487733.2020.1776561

  1. Did this theme about female care obligations not emerge from the analysis?
  2. Lines 107 to 108 mentions negative parenting behaviors. Could you please provide examples?
  3. Line 115: should be “America” instead of “American” to reflect the same in Line 120 for consistency
  4. Line 130: please indicate the exact number of participants, please? Use this for your n value.
  5. Line 175: did you use coding software e.g. NVivo?
  6. Line 190, does the quote come from a parent? Do you know their age?
  7. Line 217: please use a colon instead of a semicolon to maintain consistency with prior quotes.
  8. Line 247: please capitalize the letters O, S and E in this subtitle
  9. Line 253: you have provided the age of this child. Did you collect information about the age of the parents?
  10. Line 394: I thought you interviewed only families; this quote is apparently from a teacher.
  11. Line 410: please see my comment 3 above regarding age groups defining children and adolescents.

If you decide to incorporate these revisions, please upload a manuscript that contains tracked changes or other method to highlight revisions.  Thank you for the opportunity to review this work.

Author Response

  1. Line 16 indicates n=45 families, but how many participants were interviewed exactly?
    1. Inserted in line 151
  1. Line 38 – economic is split up into two words for some reason.
    1. Changed
  1. Line 44: what are the defined age groups of children and adolescents that you have accepted from the literature? E.g. school aged children were defined as 4 to 12 years, and adolescents as 13 to 19 years?
    1. Added a line which reads “These policies have transformed how children (defined as aged 4-12 years) and adolescents (defined as 13-18 years) behave in their daily lives”
    2. See line 47-48
  1. Line 45 mentions disruption to care arrangements as a result of the pandemic. There are also disruptions to the burden of this care, for example on women. There are also disruptions to families’ incomes, as many people have been laid off, their work hours cut, or are expected to work from home, and these are significant disruptions to the family life, and the quality of life of households.   It would have significant impacts on the health and wellbeing of children, too. It may be worthwhile to cite more information from: Syed, I.U. and Ahmad, F. (2020).  COVID19 and healthcare workers’ struggles in long term care homes. Special Issue: Health of frontline workers during the COVID-19 pandemic. The Journal of Concurrent Disorders. Available from:   https://concurrentdisorders.ca/2020/11/08/covid-19-and-healthcare-workers-struggles-in-long-term-care-homes/   
    1. Added a new section which reads “Disruptions have been shown to disproportionately effect women (Clark et al., 2020), while reductions in families’ incomes, from reduction in work hours, have negatively impacted on the quality of life of households.  These have had significant impacts on the health and wellbeing of children, too (Syed and Ahmad, 2020). See lines 50-54
  1. It may also be worthwhile to cite the work of Kate Power: The COVID-19 pandemic has increased the care burden of women and families.  Sustainability: Science, Practice and Policy, Available from: https://www.tandfonline.com/doi/full/10.1080/15487733.2020.1776561 
    1. We have included the reference ‘Clark, S., McGrane, A., Boyle, N., Joksimovic, N., Burke, L., Rock, N. and O’ Sullivan, K. (2020), ‘You’re a teacher you’re a mother, you’re a worker’: Gender inequality during Covid‐19 in Ireland.. Gender Work Organ. Accepted Author Manuscript e12611. https://doi.org/10.1111/gwao.12611

  1. Did this theme about female care obligations not emerge from the analysis?
    • This was not explored for this paper
  1. Lines 107 to 108 mentions negative parenting behaviors. Could you please provide examples?
    1. Examples have now been included in the paper see line 125
  1. Line 115: should be “America” instead of “American” to reflect the same in Line 120 for consistency
    • American has been changed to America
  1. Line 130: please indicate the exact number of participants, please? Use this for your n value
    1. Included in line 151
  1. Line 175: did you use coding software e.g. NVivo?
    1. Yes we have included this line ‘A thematic approach to IPA is used for the data analysis using the NVIVO software package.” See line 213
  1. Line 190, does the quote come from a parent? Do you know their age?
    • The quote has been updated to reflect it was a parent who made the statement. The age is unknown because parents were not asked what age they were see line 235
  1. Line 217: please use a colon instead of a semicolon to maintain consistency with prior quotes.
    • The paper now reflects a colon
  1. Line 247: please capitalize the letters O, S and E in this subtitle
    • These have now been capitalized
  1. Line 253: you have provided the age of this child. Did you collect information about the age of the parents?
    • We did not collect the age of parents. At times the age of the children came out in the interview.
  1. Line 394: I thought you interviewed only families; this quote is apparently from a teacher.
    1. This has been clarified with this line inserted “Six parents, two of whom are also special needs teachers” see line 441
  1. Line 410: please see my comment 3 above regarding age groups defining children and adolescents.
    1. Defined in the introduction line 47 and 48

Round 2

Reviewer 3 Report

All my comments have been addressed by the authors. I recommend that the manuscript should be accepted for publication.  Congratulations, and thank you for the opportunity to review your work. 

Author Response

Point 1: All my comments have been addressed by the authors. I recommend that the manuscript should be accepted for publication.  Congratulations, and thank you for the opportunity to review your work. 

Response 1: Thank you for all your comments, helping us to improve the article and recommending for publication. We look forward to having it published. 

Reviewer 4 Report

All my suggestions for revision(s) have been addressed by the authors. I recommend that the manuscript should be accepted for publication.  Congratulations, and thank you for the opportunity to review your work. 

Author Response

Point 1: All my suggestions for revision(s) have been addressed by the authors. I recommend that the manuscript should be accepted for publication.  Congratulations, and thank you for the opportunity to review your work. 

Response 1: Thank you for all your comments, helping us to improve the article and recommending for publication. We look forward to having it published.